# Gut Microbiota and Bile Acids Mediate the Clinical Benefits of YH1 in Male Patients with Type 2 Diabetes Mellitus: A Pilot Observational Study

**DOI:** 10.3390/pharmaceutics14091857

**Published:** 2022-09-02

**Authors:** Yueh-Hsiang Huang, Yi-Hong Wu, Hsiang-Yu Tang, Szu-Tah Chen, Chih-Ching Wang, Wan-Jing Ho, Yi-Hsuan Lin, Geng-Hao Liu, Pei-Yeh Lin, Chi-Jen Lo, Yuan-Ming Yeh, Mei-Ling Cheng

**Affiliations:** 1Department of Traditional Chinese Medicine, Chang Gung Memorial Hospital, Taipei 105, Taiwan; 2Graduate Institute of Clinical Medical Sciences, Chang Gung University, Taoyuan 333, Taiwan; 3School of Traditional Chinese Medicine, College of Medicine, Chang Gung University, Taoyuan 333, Taiwan; 4Metabolomics Core Laboratory, Healthy Aging Research Center, Chang Gung University, Taoyuan 333, Taiwan; 5Division of Endocrinology and Metabolism, Department of Internal Medicine, Chang Gung Memorial Hospital, Linkou Branch, Taoyuan 333, Taiwan; 6Division of Cardiology, Department of Internal Medicine, Chang Gung Memorial Hospital, Linkou Branch, Taoyuan 333, Taiwan; 7Department of Traditional Chinese Medicine, Chang Gung Memorial Hospital, Linkou Branch, Taoyuan 333, Taiwan; 8Department of Medical Nutrition Therapy, Chang Gung Memorial Hospital, Taoyuan 333, Taiwan; 9Genomic Medicine Core Laboratory, Chang Gung Memorial Hospital, Linkou Branch, Taoyuan 333, Taiwan; 10Graduate Institute of Health Industry Technology, Chang Gung University of Science and Technology, Taoyuan 333, Taiwan; 11Clinical Metabolomics Core Laboratory, Chang Gung Memorial Hospital, Taoyuan 333, Taiwan; 12Department of Biomedical Sciences, College of Medicine, Chang Gung University, Taoyuan 333, Taiwan

**Keywords:** type 2 diabetes, Chinese herbal medicine, YH1, gut microbiota, bile acids

## Abstract

Our previous clinical trial showed that a novel concentrated herbal extract formula, YH1 (*Rhizoma coptidis* and Shen-Ling-Bai-Zhu-San), improved blood glucose and lipid control. This pilot observational study investigated whether YH1 affects microbiota, plasma, and fecal bile acid (BA) compositions in ten untreated male patients with type 2 diabetes (T2D), hyperlipidemia, and a body mass index ≥ 23 kg/m^2^. Stool and plasma samples were collected for microbiome, BA, and biochemical analyses before and after 4 weeks of YH1 therapy. As previous studies found, the glycated albumin, 2-h postprandial glucose, triglycerides, total cholesterol, and low-density lipoprotein cholesterol levels were significantly improved after YH1 treatment. Gut microbiota revealed an increased abundance of the short-chain fatty acid-producing bacteria *Anaerostipes* and *Escherichia/Shigella*. Furthermore, YH1 inhibited specific phylotypes of bile salt hydrolase-expressing bacteria, including *Parabacteroides*, *Bifidobacterium*, and *Bacteroides caccae*. Stool tauro-conjugated BA levels increased after YH1 treatment. Plasma total BAs and 7α-hydroxy-4-cholesten-3-one (C4), a BA synthesis indicator, were elevated. The reduced deconjugation of BAs and increased plasma conjugated BAs, especially tauro-conjugated BAs, led to a decreased glyco- to tauro-conjugated BA ratio and reduced unconjugated secondary BAs. These results suggest that YH1 ameliorates T2D and hyperlipidemia by modulating microbiota constituents that alter fecal and plasma BA compositions and promote liver cholesterol-to-BA conversion and glucose homeostasis.

## 1. Introduction

Traditional Chinese medicine (TCM) has been an excellent resource for developing new medications for metabolic disorders, such as type 2 diabetes (T2D), since ancient times [1]. Increasing evidence indicates that TCM can treat metabolic disorders by regulating the composition of gut microbiota, such as bacteria with bile salt hydrolase (BSH) activity and short-chain fatty acid (SCFA)-producing bacteria [2]. Our previous randomized double-blind placebo-controlled pilot study discovered that YH1, an innovative antidiabetic medication, could improve hypoglycemic action, β-cell function, and lipid metabolism in overweight/obese patients with poorly controlled T2D who had taken three or more classes of oral hypoglycemic agents (OHAs) before enrollment [3]. YH1 was designed as a concentrated herbal granule containing *Rhizoma coptidis* (RC) and Shen-Ling-Bai-Zhu-San (SLBZS). Modern pharmacological research has identified that the major chemical constituents of RC are alkaloids, and berberine serves as its main bioactive alkaloid with antidiabetic, antidyslipidemic, antiobesity, antibiotic, antioxidant, and anti-inflammatory activities [4]. However, the bioavailability of berberine is far below 1% in plasma, due to poor intestinal absorption and efflux via the action of P-glycoprotein [5]. Previous studies in mice showed that RC alkaloids induced hypolipidemic effects through the modulation of gut microbiota and hepatic lipid metabolism [2,6]. In addition, recent experimental and clinical studies have reported that berberine can reduce inflammation and regulate blood sugar and lipid levels by changing the composition of gut microbes and bile acids (BAs) [2,6,7,8,9]. As shown in both in vivo and in vitro studies [7], berberine exposure changed bacterial community composition and function, particularly reducing BSH-expressing bacteria such as *Clostridium spp.* Metagenomic and metabolomic studies demonstrated that the antidiabetic effect of berberine was mediated by the inhibition of deoxycholic acid (DCA) biotransformation by *Ruminococcus bromii* [9], suggesting a potential therapeutic approach for patients with T2D through the modulation of microbiome dysbiosis. Therefore, it is critical to understand whether the composition of gut microbes and BAs of patients with T2D and hyperlipidemia are modulated by YH1 treatment.

There are bidirectional interactions between gut microbiota and BA synthesis [10]. Gut microbiota affects the BA pool size, composition, and metabolism, including deconjugation and 7α-dehydroxylation of conjugated and primary BAs, as well as the formation of secondary BAs, whereas BAs can control intestinal bacterial overgrowth and prevent inflammation [11]. In addition, BAs play important roles in the homeostasis of lipid, glucose, and energy metabolism by regulating two major BA receptors: the farnesoid X receptor (FXR) and Takeda G protein-coupled receptor 5 (TGR5) [12]. FXR is the major sensor and regulator of BA metabolism, which regulates BA pool size via a negative feedback loop involving intestinal and hepatic FXR signaling [13]. Deconjugation of glyco (G)- and tauro (T)-conjugated BAs is performed by microbial BSH. A recent study reclassified BSHs into eight phylotypes and demonstrated the variation in different microbial deconjugation abilities of glyco- and tauro-conjugated BAs [14]. Therefore, changes in gut microbiota composition affect BA metabolism and are thus associated with BA pool size alteration, the conjugated to unconjugated BA ratio, G/T-BAs ratio, and primary to secondary BA ratio. The most potent endogenous agonists for TGR5 are tauro-conjugated BAs, followed by unconjugated BAs and glyco-conjugated BAs [15]. Furthermore, the activation of TGR5 by secondary BAs is greater than that of primary BAs, so taurolithocholic acid (TLCA) is the most potent endogenous activator of TGR5 [16]. However, conjugated BAs are reportedly inactive to nuclear receptors (FXR) in the absence of a specific transporter, and the effective dose of BAs for TGR5 was lower than that for FXR [17]. Activation of TGR5 in intestinal L cells by conjugated BAs enhanced the secretion of glucagon-like peptide-1 (GLP-1), and FXR exhibited crosstalk with TGR5 to control GLP-1 secretion [11]. Overall, in addition to BA synthesis and excretion playing a critical role in affecting lipid catabolism, alteration of BA composition directly affects their signaling due to different affinities to FXR and TGR5 in the host, thus regulating lipid, glucose, and energy homeostasis [12].

SCFAs, including acetate, propionate, and butyrate, are well-known to induce GLP-1 secretion from gut L cells, and thus indirectly regulate glucose homeostasis by promoting insulin secretion and reducing pancreatic glucagon secretion [18]. SLBZS was reported to modulate the gut microbiota during the treatment of functional dyspepsia or inflammatory bowel disease by enriching SCFA-producing bacteria [19,20]. Moreover, berberine can enrich SCFA-producing bacteria, especially butyrate-producing bacteria, such as *Enterobacter* and *Escherichia/Shigella*, resulting in reduced plasma glucose and lipid levels [21]. A daily dose of YH1 contains 360.9 mg of berberine. Although the YH1 treatment used in our previous study had a four-fold lower berberine dosage than the dose of berberine used alone in another study [22], YH1 treatment achieved an 11.1% better reduction in glycated hemoglobin (HbA1c) levels, along with larger, favorable decreases in serum triglycerides (TG), total cholesterol, and low-density lipoprotein cholesterol (LDL-C) levels [3]. This suggests that YH1 treatment has superior antidiabetic and antidyslipidemic activities compared with using the single pure compound alone. To evaluate the mechanisms of YH1-mediated hypoglycemic and hypolipidemic effects, we investigated alterations in the stool microbiome, as well as in BA composition of plasma and fecal samples from patients with T2D and hyperlipidemia following 4 weeks of YH1 treatment.

## 2. Materials and Methods

### 2.1. Study Design and Participants

We conducted this pilot observational study in the TCM clinics of the Taoyuan branches of Chang Gung Memorial Hospital from January 2020 to December 2021 (ClinicalTrials.gov number, NCT04194515). This study was approved by the Committee on Research Ethics of the Chang Gung Memorial Hospital in Taiwan (No: 201901022B0A3). All participants provided written informed consent prior to participation. Eligible participants were male outpatients with T2D and hyperlipidemia who were not receiving treatment and consented to taking 6 g of YH1 [3] three times daily for 4 weeks. Patients were eligible to be enrolled in this study only if they met all of the following criteria: (1) Male patients with T2D who did not take hypoglycemic agents in the past month; (2) Aged 20–65 years; (3) Body mass index (BMI) ≥ 23 kg/m^2^; (4) Received dietary and exercise education from a nutritionist for at least one month; (5) HbA1c ≥ 6.5%; (6) LDL-C ≥ 130 mg/dl; (7) Agreed to take 6 g of YH1 three times per day for 4 weeks.

Patients were excluded if they met any of the following criteria: (1) Type 1 or other specific type of diabetes; (2) Female sex; (3) Use of oral hypoglycemic agents or insulin within the past month; (4) Use of lipid-lowering agents within the past month; (5) Serious gastrointestinal (GI) tract diseases, such as peptic ulcers or GI tract bleeding; (6) Hepatic insufficiency with alanine aminotransferase (ALT) > 108 U/L or renal insufficiency with estimated glomerular filtration rate (eGFR) < 60 mL/min/1.73 m^2^; (7) stressful situations, including diabetic ketoacidosis, nonketotic hyperosmolar diabetic coma, severe infection, or surgery in the previous month; (8) Mental illness; (9) Addiction to tobacco, alcohol or other drugs; (10) Hemoglobin-related disease or chronic anemia; (11) Underlying conditions that could lead to poor compliance; (12) Severe organ disease, including cancer, coronary artery disease, myocardial infarction, or cerebrovascular diseases; (13) Consecutive use of antibiotics, probiotics, or weight loss drugs for more than 3 days within the 3 months prior to participation in this study; (14) Uncontrolled hypertension (blood pressure ≥ 160/100 mmHg); (15) Chinese medicine treatment in the past month.

All participants were instructed to maintain their eating habits and lifestyles during the study period, and dietary recalls were obtained to document usual dietary intake by the same nutritionist before and 4 weeks after YH1 treatment. Patients were evaluated for medication history, Bristol Stool Form Scale (BSFS), Constipation Assessment Scale (CAS), prescription adherence to YH1, and adverse events at 0 and 4 weeks. Additionally, physical examination parameters, including body weight, BMI, waist circumference, blood pressure, and heart rate, were measured. Fecal specimens were collected at home, with a clean stool collection container with a spoon attached to the lid, prior to and upon conclusion of YH1 treatment. Stool samples were stored in the freezer immediately after collection, transferred to the laboratory in the frozen state within 3 days, and stored at −80 °C before analysis. In addition, blood samples were collected while fasting and 2 h post-breakfast for assessment of metabolic and biochemical parameters before and 4 weeks after YH1 treatment. A standardized breakfast containing a Chinese omelet with pork chop and a slice of pan-fried radish cake (total 440 calories, including 45 g carbohydrates, 20 g protein, and 20 g fat) was provided at 0 and 4 weeks to avoid the effect of different foods on BA profiles. 

### 2.2. Microbiome, Metabolomic, and Biochemical Measures

For stool microbiome analysis, we conducted DNA extraction, polymerase chain reaction (PCR) amplification, and 16S rRNA sequencing, as previously described [23]. Total bacterial genomic DNA was isolated from fecal specimens using a QIAamp PowerFecal^®^ DNA Kit (Qiagen, USA) in accordance with the manufacturer’s protocol. Genomic DNA was amplified by PCR with primers targeting the 16S rRNA V3-V4 region, with a product size of 460 bp. The detailed protocol for PCR and sequencing has been described in a previous study [24]. Briefly, adapter overhang nucleotide sequences (Illumina Inc) were added to the gene-specific sequences (forward primer sequence, 5′-TCGTCGGCAGCGTCAGATGTGTATAAGAGACAGCCTACGGGNGGCWGCAG-3′, and reverse primer sequence, 5′-GTCTCGTGGGCTCGGAGATGTGTATAAGAGACAGGACTACHVGGGTATCTAATCC-3′). Subsequently, amplicon products were purified using Agencourt AMPure XP beads (Beckman Coulter), followed by indexing and barcoding PCR. The final amplicon libraries were approximately 630 bp, and the Agilent 2100 Bioanalyzer with the Agilent HS DNA Kit was used for size and quality validation. Multiplexed, pooled library sequencing was conducted on the MiSeq System with MiSeq Reagent Kit v3 (600-cycle) (Illumina, San Diego, CA, USA). Microbiome analysis was executed by USEARCH (v11), following the methods described in a previous study [24]. Subsequent analysis was performed only for sequence tags ≥400 bp in length. The operational taxonomic units (OTUs) were clustered by the UPARSE [25] algorithm. In addition, the RDP training set (v16) was used as a database of reference species, and the SINTAX algorithm was performed for final taxonomic classification [26]. Measurements of α-diversity (Chao1 and Shannon index) and β-diversity (Bray–Curtis dissimilarity) were assessed by QIIME (version 1.9.1) [27] via DESeq2 [28] normalization.

Fasting plasma glucose (FPG), 2-h postprandial glucose (2hPG), and glycated albumin were measured at weeks 0 and 4. Homeostatic model assessment of insulin resistance (HOMA-IR), β-cell function index (HOMA-β), lipid profiles, C-reactive protein (CRP), as well as hepatic and renal function were assessed at the beginning and end of the study. Furthermore, 2-h postprandial blood samples were collected for measurement of 2hPG and stored at −80 °C until analyses of plasma BA profiles, 7α-hydroxy-4-cholesten-3-one (C4), and fibroblast growth factor 19 (FGF19), before and after YH1 therapy. C4 and FGF19 serum biomarkers were indicators of BA synthesis and an index of the negative feedback loop involved in BA metabolism, respectively.

Plasma and fecal sample preparation and analysis of BA by a UPLC–MS/MS system (Waters, Milford, CT, USA) covering 15 major BA species are described in Protocol S1. The full term of each BA and its abbreviation are presented in the table of Protocol S1. We assessed the total and individual concentrations of the 15 BAs, as well as the ratios of classified BAs, including ratios of 12α-hydroxylated BAs (CA, GCA, TCA, DCA, GDCA, and TDCA) to non-12α-hydroxylated BAs (CDCA, GCDCA, TCDCA, LCA, GLCA, TLCA, UDCA, GUDCA, and TUDCA); primary (CA, GCA, TCA, CDCA, GCDCA, TCDCA) and secondary BAs (DCA, GDCA, TDCA, LCA, GLCA, TLCA, UDCA, GDCA, TUDCA); and glyco- plus tauro-conjugated (GCA, GCDCA, GDCA, GLCA, GUDCA, TCA, TCDCA, TDCA, TLCA, TUDCA) to unconjugated BAs (CA, CDCA, DCA, LCA, UDCA).

For plasma C4 analysis, 300 μL methanol was added to 100 μL plasma samples for liquid–liquid extraction. Samples of the mixture were incubated on ice for 30 min and then centrifuged to precipitate protein at 12,000 rpm for 30 min at 4 °C. The supernatant was transferred to a sample vial and analyzed in an LC–MS system (UPLC with Xevo TQS MS, Waters, Manchester, UK) in negative atmospheric pressure chemical ionization (APCI) mode with multiple reaction monitoring. The chromatographic separation was achieved on an Acquity HSS pentafluorophenyl (PFP) column (2.1 × 100 mm, particle size of 1.8 μm, Waters Corp., Milford, CT, USA) at 25 °C with mobile phase A (25% acetonitrile with 0.1% formic acid) and mobile phase B (methanol). The flow rate was set to 0.3 mL/min. The gradient profile was as follows: 70% B for 1 min; linear gradient 70–75.7% B, 8 min; 75.7–100% B, 1 min; and 100% B, 2.2 min. The column was then re-equilibrated for 3 min. The parameters of MS were as follows: corona was 1 μA, desolvation gas flow was 800 L/h at 600°C, and cone gas flow was 150 L/h at 150 °C. System operation and data acquisition were controlled using Mass Lynx software and targeted metabolic data were analyzed by TargetLynx (Waters, Milford, CT, USA). In addition, serum FGF19 was measured by using commercially available enzyme-linked immunosorbent assay kits (R&D Systems, Minneapolis, MN, USA) according to the manufacturer’s instructions.

### 2.3. Statistical Analysis

IBM SPSS statistical software (Version 21, Armonk, NY, USA) was used for nonparametric analysis and verification. Measurement data are presented as the median (minimum, maximum), and categorical data are described by numbers. The Wilcoxon signed-rank test was used to detect differences in gut microbial features (richness, diversity, and relative abundances of species), plasma and fecal BA levels, and clinical outcomes between baseline and post-treatment measurements. Fisher’s exact test was performed to assess categorical variables. Spearman’s test was used for correlation analyses, not only between the percentage change in C4 and FGF19, but also between the gut microbiome and host metabolome or clinical parameters. All statistics were performed using a two-tailed test. Values of *p* < 0.05 (*) were considered statistically significant, and *p* < 0.01 (**) was considered highly significant.

## 3. Results

### 3.1. Patient Characteristics

Ten male patients with hyperglycemia and hyperlipidemia who agreed to take 6 g of YH1 three times per day for 4 weeks were recruited for the study, and no participants dropped out. Prescription adherence of all patients was above 80%, and no participant was excluded from the data analysis. Participants were aged 38 to 57 years, with a median age of 48 years. The median duration of diabetes was 2.0 years, with a range from 0.5 to 14.0 years. Previous medications prescribed to treat chronic diseases included hypouricemic agents (sulfinpyrazone and benzbromarone) taken by two patients for a long time, four antihypertensive drugs taken by three participants (one individual received Exforge, one used olmesartan, and one took both Sevikar and lercanidipine), antihistamines (levocetirizine and denosin) used by two people, and silymarin used by two individuals. Participants continued to take the above medication without dosage or medication changes during the study.

### 3.2. Microbiota Profiles

Stool samples were collected from the participants before and after 4 weeks of YH1 treatment. Microbiota profiles were investigated using 16S rRNA sequencing. Microbial community variation, as indicated by Chao1 and Shannon indices (α-diversity), was slightly decreased after YH1 treatment without a significant difference (Figure 1a). Principal coordinate analysis (PCoA) revealed that the β-diversity of the gut microbiota at baseline and after treatment did not show significant changes (Figure 1b). Taxonomic profiles of the bacterial community at the genus level revealed elevated abundances of *Anaerostipes* (0.09 to 0.56%, *p* = 0.015) and *Escherichia/Shigella* (2.22 to 10.18%, *p* < 0.005), but reduced abundances of *Parabacteroides* (1.88 to 0.52%, *p* < 0.005), *Bifidobacterium* (1.31 to 0.09%, *p* = 0.036), and *Romboutsia* (0.25 to 0.01%, *p* < 0.005), after YH1 therapy (Figure 1c). YH1 reduced the abundance of bacteria species (Figure 1d) including *Parabacteroides distasonis* (0.92 to 0.32%, *p* = 0.043), *Parabacteroides merdae* (0.84 to 0.14%, *p* = 0.031), *Oscillibacter ruminantium* (0.39 to 0.13%, *p* = 0.031), and *Bacteroides caccae* (0.91 to 0.30%, *p*
*=* 0.011). Collectively, our microbiome analysis indicated that, although YH1 did not significantly alter the richness and diversity of gut microbiota, specific gut bacteria showed prominently different abundances at the genus and species levels after 4 weeks of YH1 treatment. YH1 enriched SCFA-producing bacteria, including *Anaerostipes* and *Escherichia/Shigella*, and inhibited BSH-active clones, such as *Parabacteroides*, *Bifidobacterium*, and *Bacteroides caccae*.

### 3.3. Changes in Plasma and Fecal Bile Acid Profiles, Plasma C4, and FGF19 Levels

In this study, plasma and stool samples were collected from ten patients at baseline and 4 weeks after oral administration of YH1. The changes in 15 types of plasma BAs are shown in Figure 2. Plasma levels of glyco- and tauro-conjugated primary BAs (GCA, TCA, GCDCA, TCDCA) all significantly increased. The concentrations of three other tauro-conjugated secondary BAs (TDCA, TLCA, TUDCA) were also elevated. Regarding glyco-conjugated secondary BAs, only levels of GDCA significantly increased. However, two unconjugated secondary BAs (DCA and LCA) showed significantly reduced concentrations. Regarding the change in the percentage composition of total BAs after YH1 treatment (Appendix A), the median percentage of CAs (sum of CA, GCA, and TCA) significantly increased from 10.3 to 15.9% (*p*
*=* 0.013). The median percentage of CDCAs (sum of CDCA, GCDCA, and TCDCA) also increased prominently from 41.9 to 56.7% (*p =* 0.007). However, the median percentage of secondary BAs, including DCAs (sum of DCA, GDCA, and TDCA), LCAs (sum of LCA, GLCA, and TLCA), and UDCAs (sum of UDCA, GUDCA, and TUDCA), significantly decreased from 33.7 to 24.1% (*p*
*=* 0.007), 2.3 to 1.5% (*p*
*=* 0.037), and 6.9 to 3.2% (*p*
*=* 0.005), respectively. Therefore, the ratios of CAs/DCAs and CDCAs/DCAs both increased significantly after YH1 treatment, indicating that YH1 reduced the production of secondary BAs.

Regarding the change in BA classification after YH1 treatment from baseline (Figure 3), the levels of 12α-hydroxylated BAs increased after YH1 treatment (*p*
*=* 0.047), with the median concentration increasing from 827.1 to 1073.2 nM. Similarly, the median concentration of non-12α-hydroxylated BAs significantly increased from 1036.9 to 2037.8 nM at the end of YH1 treatment (*p*
*=* 0.005). Levels of primary BAs in plasma increased significantly (*p*
*=* 0.005), with the median concentration increasing from 1118.4 to 2354.2 nM, whereas levels of secondary BAs did not change. Conjugated BA levels were significantly elevated (*p*
*=* 0.005), with a median increase from 1261.4 to 2872.0 nM. Interestingly, levels of both glyco- and tauro-conjugated BAs were strikingly elevated, and the increase in tauro-conjugated BA levels was significantly greater than that of glyco-conjugated BA levels after YH1 treatment. Therefore, the median ratio of G/T-BAs decreased from 10.7 to 4.2 (*p*
*=* 0.005), and a decline in the G/T ratio was observed in both primary and secondary BAs. Furthermore, the median unconjugated BA level decreased from 545.0 to 289.2 after treatment (*p*
*=* 0.047). The ratio of different classifications of plasma BAs before and after YH1 treatment are presented in Figure 4. The ratio of 12α-hydroxylated to non-12α-hydroxylated BAs showed no significant change. The ratio of primary to secondary BAs increased significantly, with a median increase from 1.4- to 2.5-fold (*p*
*=* 0.005). The ratio of conjugated to unconjugated BAs also increased prominently, with a median increase from 2.0- to 7.0-fold (*p*
*=* 0.007), suggesting that YH1 inhibited the deconjugation of conjugated BAs by modulating the gut microbiome. 

Figure 4 shows that the total levels of plasma BAs increased significantly after YH1 treatment (*p* = 0.017), with the median concentration increasing from 1914.5 to 3200.5 nM. C4 levels were also markedly elevated (*p*
*=* 0.037), but plasma FGF19 levels showed no significant change before and after YH1 treatment (Appendix A). However, there was a significant negative correlation between the percentage change in C4 and FGF19 (r = −0.7; *p* = 0.025). Intestinal FXR regulated the production of FGF19, a hormone that travels via enterohepatic circulation with a negative feedback effect of hepatic CYP7A1 activity on BA and lipid metabolism. A lower FGF19 concentration suggested less activation of intestinal FXR signaling, thus leading to increases in BA biosynthesis and plasma C4 levels after YH1 treatment.

The analysis of BAs in fecal samples indicated that concentrations of stool TCDCA, TDCA, and TLCA were significantly increased after YH1 therapy (Appendix A). The total levels of tauro-conjugated BAs in feces increased significantly, with the median concentration rising from 363.2 to 747.3 nmol/mg after treatment (*p*
*=* 0.047). However, there were no significant differences in total stool BAs or the ratios of different classifications in feces (Figure 4). Therefore, clinical YH1 treatment was found to be related to prominent modulation of plasma BA profiles and elevated levels of the fecal tauro-conjugated BAs in male individuals with T2D and hyperlipidemia.

### 3.4. Changes in Clinical Parameters and the Correlation between YH1-Responsive Microbiota and BA or Clinical Outcomes

Changes in various clinical parameters before and after four weeks of YH1 treatment are shown in Table 1. Regarding anthropometric characteristics, there were no significant differences in body weight, waist circumference, blood pressure, or heart rate. After YH1 treatment, the median glycated albumin level significantly decreased from 16.8 to 15.3% (*p*
*=* 0.005), without hypoglycemic episodes. The median 2hPG level also declined from 158.0 to 117.5 mg/d (*p*
*=* 0.007). However, no significant changes were found in FPG, HOMA-IR, or HOMA-β scores at the end of this study. Regarding the parameters of lipid metabolism, the 4-week YH1 treatment resulted in significant reductions (*p* < 0.01) in total cholesterol, LDL-C, and TG levels compared with the baseline values (Table 1). Although high-density lipoprotein cholesterol (HDL-C) levels decreased synchronously, the TG/HDL-C value, a predictor of cardiovascular disease, also decreased significantly after the treatment, with a median reduction from 4.4 to 2.7. Levels of the liver enzyme alanine aminotransferase (ALT) showed a decreasing trend after YH1 treatment (*p*
*=* 0.07), with a median decrease from 49.5 to 39.5 U/L. However, neither inflammatory marker (CRP) levels nor renal function was significantly different after YH1 treatment in this study. Regarding the assessment of bowel movements, there were no significant changes in CAS or BSFS scores during this study. Adverse GI events were found once in two patients, experiencing mild nausea and bloating, respectively. The remaining eight patients did not experience any side effects during this trial. The severity of the above GI events was below grade two, as measured by the Common Terminology Criteria for Adverse Events (CTCAE, Version 4.0) grading system. YH1 treatment was temporarily paused for one day when the two participants reported discomfort, and treatment was resumed once symptoms were resolved without other medical intervention. No patients experienced hypoglycemia during this study.

There were some correlations between YH1-responsive microbiota and BAs or clinical outcomes. At the genus level, the increase in *Escherichia/Shigella* abundance represented a positive correlation with elevated TLCA (r = 0.648; *p* = 0.043) in plasma. *Parabacteroides* was negatively correlated with plasma TCDCA (r = −0.685; *p* = 0.029) and tauro-conjugated BAs (r = −0.661; *p* = 0.038), but positively correlated with LDL-C levels (r = 0.685; *p* = 0.029). In addition, *Bifidobacterium* was negatively correlated with TDCA (r = −0.697; *p* = 0.025) and tauro-conjugated secondary BAs (r = −0.648; *p* = 0.043) in plasma. Therefore, a decrease in *Parabacteroides* and *Bifidobacterium* abundances led to an increase in plasma levels of tauro-conjugated BAs and a decrease in LDL-C levels. At the species level, the reduction in *Parabacteroides distasonis* abundance also showed a positive correlation with decreased LDL-C levels (r = −0.782; *p* = 0.008), and the decrease in *Parabacteroides merdae* abundance was positively correlated with decreased plasma unconjugated BA levels (r = 0.745; *p* = 0.013). The decrease in *Oscillibacter ruminantium* abundance was positively correlated with reduced FPG levels (r = 0.867; *p* = 0.001), and negatively correlated with both elevated HOMA-β scores (r = −0.818; *p* = 0.004) and stool TLCA levels (r = −0.770; *p* = 0.009). In brief, YH1 significantly inhibited some microbiota in the gut, especially *Parabacteroides* and *Bifidobacterium*, which were related to the changes in BA composition, as well as in the clinical parameters.

## 4. Discussion

This pilot observational clinical study is the first to report that 4 weeks of oral YH1 treatment alone significantly alters the stool microbiome and increases levels of conjugated BAs, especially tauro-conjugated BAs, in plasma and stool. These changes could effectively improve glycemic levels and lipid profiles for middle-aged, T2D male patients with LDL-C ≥ 130 mg/dl and BMI ≥ 23 kg/m^2^. YH1 increased the abundances of SCFA-producing bacteria *Anaerostipes* and *Escherichia/Shigella* and reduced the abundances of *Parabacteroides*, *Bifidobacterium*, *Romboutsia*, *Oscillibacter ruminantium*, and *Bacteroides caccae*. Similarly, metformin improved T2D by modulating the microbiota, such as increasing the abundance of *Escherichia/Shigella* and decreasing that of *Romboutsia*, *Oscillibacter*, *Bacteroides*, and *Parabacteroides* [29]. Other antidiabetic drugs, such as DPP-4 inhibitors, reduce *Oscillibacter* abundance, and GLP-1 receptor agonist increases *Anaerostipes* abundance [29]. Therefore, YH1 could serve as an add-on medication to enhance the hypoglycemic effects of OHAs to achieve better glycemic control [3], by potentially modulating gut microbiota in patients with T2D.

Although YH1 contained berberine, the changes in microbiota and plasma BAs after YH1 treatment were partially different from those observed after a single berberine compound treatment in previous studies [7,9,30]. Previous experimental research indicated that oral administration of berberine could alter the composition of BAs by affecting gut microbes via a reduction in or elimination of *Clostridium* spp. Therefore, BSH activity was decreased, leading to an accumulation of TCA in the intestine, thus acting on FXR receptors and affecting lipid metabolism [7,30]. A clinical study reported that the hypoglycemic effect of berberine was mediated by inhibiting *Ruminococcus bromii* to attenuate DCA biotransformation [9]. In addition, berberine also depleted species including *Parabacteroides merdae* and *Bifidobacterium* spp., but enriched species such as *Parabacteroides distasonis*. Our study proved that YH1 decreased BSH-expressing microbiota, such as *Bifidobacterium*, *Parabacteroides merdae*, *Parabacteroides distasonis*, and *Bacteroides caccae*, elevating the ratio of conjugated to unconjugated BAs. Interestingly, we found that YH1 treatment decreased the G/T ratio of BAs, which was not reported in a previous clinical trial [9]. Gut microbiota with different phylotypes of BSH display some degree of selectivity for conjugated BA substrates. The phylotypes of BSH-T5 and BSH-T6 mainly comprised *Bacteroides* and *Parabacteroides*, exhibiting different deconjugating activity between tauro- and glyco-conjugated BAs [14]. Similarly, other studies suggested that the deconjugation activities of some bacteria, such as *Bacteroidetes*, was more selective for tauro-conjugated BAs than glyco-conjugated BAs [31,32]. Therefore, YH1 increased the levels of conjugated BAs, especially tauro-conjugated BAs, probably by inhibiting specific BSH-phylotype bacteria, including *Parabacteroides* and *Bacteroides caccae*. Furthermore, YH1 treatment increased the ratio of primary to secondary BAs in plasma, but the levels of primary unconjugated BAs (CA and CDCA) were not elevated. As the abundances of 7α-dehydroxylated bacteria, such as *Clostridium* (clusters XIVa and XI) and *Eubacterium*, did not show significant changes in this study and these bacteria can metabolize only unconjugated-BAs [11], the reduced production of secondary BAs (DCA and LCA) by YH1 might be related to the decrease in unconjugated BAs. Interestingly, secondary BAs and SCFAs are two major types of gut bacterial metabolites and have opposing effects on colonic inflammation. Secondary BAs, including DCA and LCA, are risk factors for gut inflammation and cancer, whereas an increase in SCFA production is associated with anti-inflammation and anticancer effects [33]. Our study demonstrated that YH1 treatment not only significantly decreased the levels of unconjugated secondary BAs, but also considerably elevated the abundances of SCFA-producing bacteria, which may play a role in protection from gut inflammation. Overall, YH1 differed from berberine in modulating gut microbiota and BA composition, which may help explain the more potent hypoglycemic and hypolipidemic effects of this herbal formula than that of the pure compound.

BAs regulate glucose, lipid, and energy homeostasis mainly through nuclear FXR and TGR5, and have pathophysiologic roles in obesity, T2D, dyslipidemia, and nonalcoholic steatohepatitis [34,35]. In addition, tauro- or glyco-conjugated forms of BAs showed agonistic activity on TGR5, with taurine conjugates being more potent than glycine conjugates [17]. In this study, YH1 treatment increased the stool levels of tauro-conjugated BAs, which can activate TGR5 in gut L cells and lead to GLP-1 secretion and glucose homeostasis. A previous research showed that berberine-enriched SCFA-producing bacteria, especially butyrate-producing bacteria, could have beneficial effects on the host [21]. Our study shows that YH1 increases the abundance of *Anaerostipes*, as well as *Escherichia/Shigella*, and both are butyrate-producing strains in the intestine. Through SCFA receptors, gut microbiota-derived SCFAs have roles in GLP-1 release, insulin secretion, and regulation of energy expenditure [18]. Collectively, the findings of this study suggest that YH1 increases not only stool tauro-conjugated BA levels, but also *Anaerostipes* and *Escherichia/Shigella*-derived SCFA levels that target receptors in gut L cells. This results in improvements in GLP-1 secretion and hyperglycemia, without any side effects of hypoglycemia. Conjugated BAs were associated with less activation of intestinal FXR signaling, so less negative feedback regulation promoted CYP7A1 activity with elevation of plasma C4 levels in our study. This led to liver cholesterol-to-BA conversion and hyperlipidemia reduction. Hence, this clinical research on YH1 could support the therapeutic mechanisms of TCM in improving metabolic diseases by modulating the gut microbiota and BAs found in animal studies [2].

Comparing our previous clinical trial [3] with the current study, YH1 not only was effective in patients with treatment-resistant T2D, but also had beneficial outcomes for drug-naïve T2D patients. Similarly, the incidence of adverse GI events, including nausea or bloating, in the YH1 group of previous trial and current study were 21.7 and 20%, respectively. In contrast to the hypoglycemic effects of OHAs alone, hyperglycemia and hyperlipidemia were significantly improved after YH1 treatment in both studies. There was no significant difference in the change of body weight or waist circumference after 4 weeks of oral YH1 treatment, possibly due to the shorter treatment period of this study. Although HDL-C was slightly reduced at week 4, the TG/HDL-C ratio decreased significantly after YH1 treatment in this study (*p* = 0.009), with a median decrease from 4.4 to 2.7. According to the previous literature, a TG/HDL-C ratio > 3.5 is linked with higher risks of cardiovascular disease and coronary heart disease-related death. It is anticipated that the future annual incidence of diabetes will increase two-fold for individuals with a TG/HDL-C ratio > 3.5 [36,37]. Nine out of ten YH1 subjects recruited for this trial met the diagnostic criteria for metabolic syndrome, which was associated with a two-fold increase in the risk of stroke and heart disease in the future compared to that in the normal population [38,39]. This study suggests that YH1 treatment improves glycemic and lipid profiles, which could result in the reduced risk of patients developing cardiovascular disease.

This study has the following limitations. First, the sample size was small, and participation in our study was limited to males only. Females were excluded to avoid blood contamination of fecal samples during the menstrual period and to avoid sex differences in the microbiota and BA compositions. The standardized breakfast was provided in this study to prevent different diet-related BA secretion, and dietary records of patients were monitored by the same nutritionist during the study. Therefore, we controlled and minimized variables that could affect gut microbiota and BA composition to make our findings more reliable. Larger randomized controlled trials are warranted to confirm these pilot results. Future research that enrolls women together with men could determine if the influences of YH1 on gut microbiota and BA are universal or gender specific for patients with T2D and hyperlipidemia. Second, this study enrolled middle-aged, overweight or obese patients with treatment-naïve T2D and hyperlipidemia in Taiwan. The results may not be applicable to individuals in different stages of T2D or other racial/ethnic populations. Finally, the 4-week study period was relatively short. Longer treatment periods are required to evaluate the long-term effects of YH1 on microbiota and BA metabolism. Notwithstanding these limitations, our study still provides important evidence that YH1 can manage T2D and hyperlipidemia by regulating the microbiome and BA composition.

## 5. Conclusions

We found that YH1 improved T2D and hyperlipidemia, by potentially inhibiting some gut bacteria with specific phenotypes of BSH, resulting in reduced deconjugation abilities. This led to elevated levels of conjugated BAs, especially tauro-conjugated BAs in plasma and stool. Consequently, YH1 potentially inhibited the activity of intestinal FXR, thereby decreasing hyperlipidemia by reducing negative feedback regulation and promoting the conversion of hepatic cholesterol to BAs. In addition, increased levels of conjugated BAs plus *Anaerostipes* and *Escherichia/Shigella*-derived SCFAs might further improve hyperglycemia by regulating the release of GLP-1 in gut L cells. The detailed and complicated mechanism of YH1 requires more clinical and basic research in the future.

## Figures and Tables

**Figure 1 pharmaceutics-14-01857-f001:**
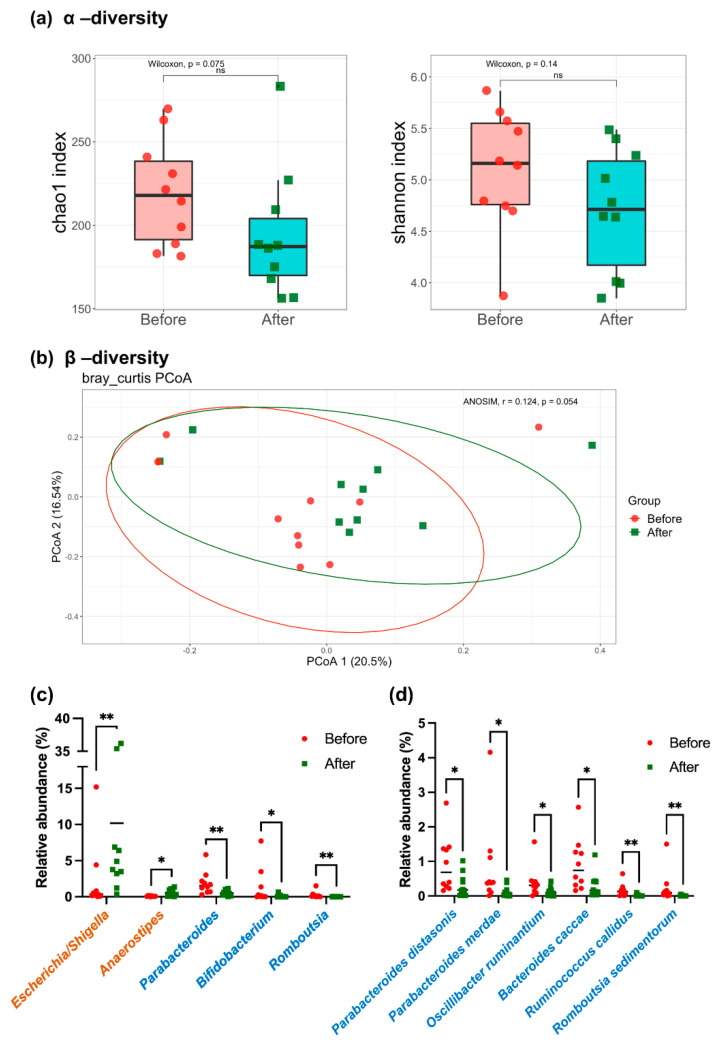
Oral YH1 modulated the composition of gut microbiota in men with type 2 diabetes and hyperlipidemia. (**a**) Chao1 and Shannon indices were used to analyze the α-diversity of fecal microbiota before (red circles) and after (green squares) YH1 treatment in ten patients. (**b**) The β-diversity of fecal microbiota at baseline and after YH1 treatment was determined by principal coordinate analysis (PCoA) based on the Bray-Curtis dissimilarity index. Significant changes in the fecal microbiota at the (**c**) genus and (**d**) species level before and after YH1 treatment are presented. Genus and species names in brown or blue font represent significant enrichment or depletion of bacteria after YH1 treatments, respectively. Statistical significance (* *p* < 0.05; ** *p* < 0.01) were detected by the Wilcoxon signed-rank test.

**Figure 2 pharmaceutics-14-01857-f002:**
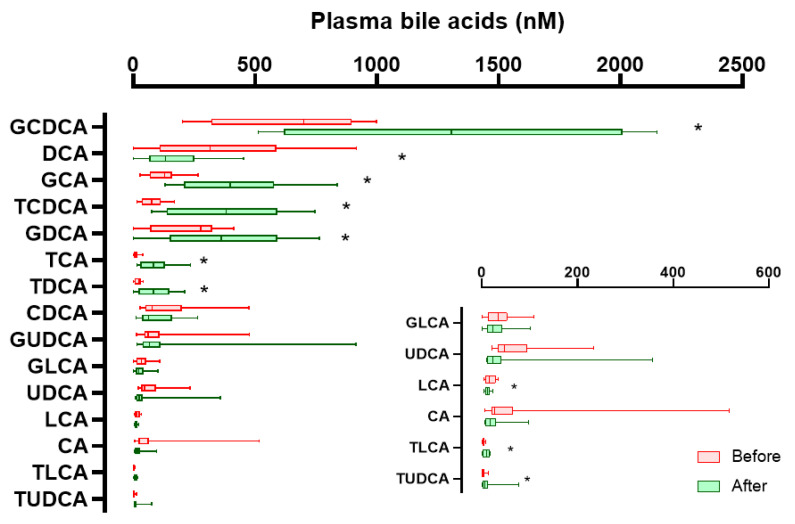
YH1 altered plasma bile acid profiles in men with type 2 diabetes and hyperlipidemia. The concentration of each bile acid in the plasma of ten patients before (red) and after (green) YH1 treatment is represented by a box-and-whisker plot. The line inside the box represents the median value, and the box is between the first and third quartiles. The ends of the whiskers represent the minimum and maximum values. Wilcoxon signed-rank test was used, and * *p* < 0.05 was considered statistically significant.

**Figure 3 pharmaceutics-14-01857-f003:**
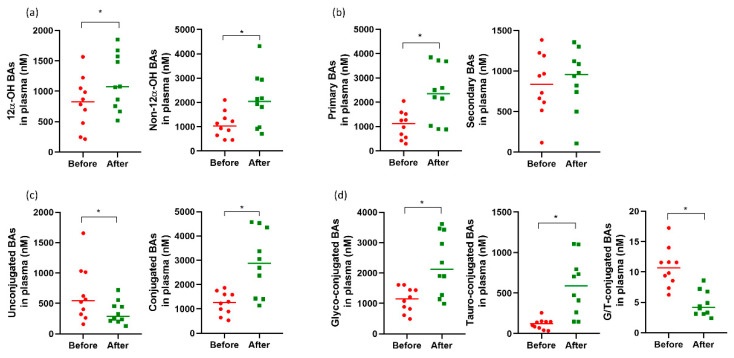
Changes in the composition of bile acids (BAs) of various categories in plasma before and after treatment with YH1. The concentrations of classified BAs in plasma before (red circles) and after (green squares) YH1 treatment in the ten patients are shown as scatter plots, with lines representing median values. Statistical differences (* *p* < 0.05) were detected by the Wilcoxon signed-rank test. After four weeks of YH1 treatment, the levels of (**a**) 12α-hydroxylated BAs and non-12α-hydroxylated BAs, (**b**) primary BAs, (**c**) conjugated BAs, and (**d**) glyco (G)- and tauro (T)-conjugated BAs in plasma were all significantly elevated. (**c**) Unconjugated BAs and (**d**) G/T ratio of conjugated BAs were significantly reduced. 12α-OH BAs, 12α-hydroxylated bile acids; Non-12α-OH BAs, non-12α-hydroxylated bile acids.

**Figure 4 pharmaceutics-14-01857-f004:**
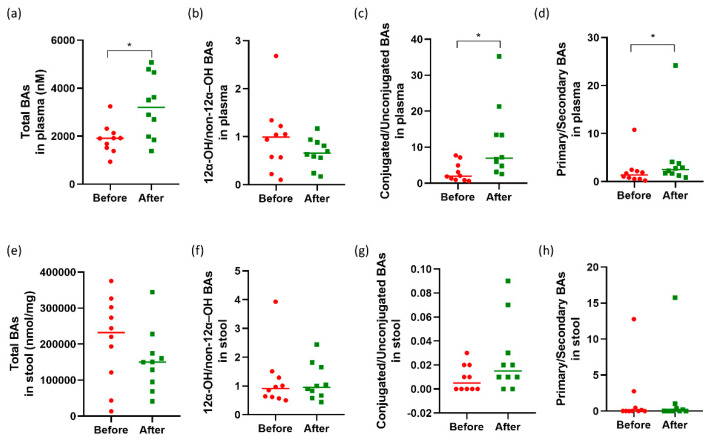
Changes in total and classification ratios of bile acids (BAs) in plasma and feces after YH1 treatment. Total BAs and classification ratios of BAs in plasma and feces before (red circles) and after (green squares) YH1 treatment in the ten patients are shown as scatter plots. The line in the graph represents the median value. The Wilcoxon signed-rank test was used to determine whether there was a statistical difference before and after treatment (* *p* < 0.05). (**a**) Total plasma BAs increased significantly after YH1 treatment. (**b**) The ratio of 12α-hydroxylated to non-12α-hydroxylated BAs didn’t show significant change. Changes in the ratio of BA classification were only significantly elevated in plasma (**c**) conjugated/unconjugated BAs and (**d**) primary/secondary BAs. (**e**) Total stool BAs had no significant change before and after YH1 treatment. In addition, no significant difference was found in the analysis of fecal BA profiles (**f**–**h**) before and after treatment. 12α-OH/non-12α-OH BAs, 12α-hydroxylated/non-12α-hydroxylated bile acids.

**Table 1 pharmaceutics-14-01857-t001:** Clinical parameters before and after YH1 treatment.

	Pre-Treatment (*n* = 10)	Post-Treatment (*n* = 10)	*p* Value
	Median (Min, Max)	Median (Min, Max)	
Demographic characteristics			
Age (years)	48 (38, 57)		
Gender, Male/Female	10/0		
Duration of DM (years)	2.0 (0.5, 14.0)		
**Anthropometric characteristics and vital signs**			
Weight (kg)	78.8 (68.4, 102.6)	79.5 (68.2, 101.8)	0.76
BMI (kg/m^2^)	27.2 (24.8, 34.2)	27.4 (24.7, 33.3)	0.77
SBP (mmHg)	133.0 (104.0, 153.0)	128.5 (103.0, 150.0)	0.42
DBP (mmHg)	86.0 (51.0, 101.0)	77.0 (62.0, 104.0)	0.17
HR (beat/min)	76.0 (60.0, 89.0)	73.0 (56.0, 80.0)	0.18
**Laboratory data**			
Glycated albumin (%)	16.8 (14.0, 32.1)	15.3 (12.8, 29.8)	0.005 **
FPG (mg/dL)	129.5 (81.0, 277.0)	130.0 (86.0, 255.0)	0.21
2hPG (mg/dL)	158.0 (111.0, 335.0)	117.5 (98.0, 304.0)	0.007 **
ALT (U/L)	49.5 (21.0, 69.0)	39.5 (12.0, 74.0)	0.07
Cr (mg/dL)	0.9 (0.5, 0.9)	0.7 (0.6, 1.0)	0.10
Total cholesterol(mg/dL)	231.5 (212.0, 262.0)	193.0 (141.0, 223.0)	0.007 **
HDL-C (mg/dL)	43.5 (34.0, 61.0)	40.0 (29.0, 50.0)	0.02 *
LDL-C (mg/dL)	150.0 (139.0, 196.0)	134.5 (88.0, 150.0)	0.007 **
Triglycerides (mg/dL)	194.0 (74.0, 497.0)	103.5 (58.0, 211.0)	0.005 **
Triglycerides/HDL-C	4.4 (1.2, 11.8)	2.7 (1.2, 4.9)	0.009 **
Fasting insulin (μU/mL)	10.2 (5.7, 20.1)	11.0 (5.3, 26.6)	0.14
HOMA-IR	3.8 (2.1, 7.7)	3.5 (2.0, 10.1)	0.24
HOMA-β	55.3 (13.0, 402.0)	81.9 (16.9, 275.1)	0.14
CRP (mg/L)	2.5 (0.4, 4.8) ^#^	1.9 (0.6, 13.5) ^#^	0.95

^#^ Only nine subjects were analyzed because of missing data. The data are presented as medians (min, max). Statistical significance (* *p* < 0.05; ** *p* < 0.01) was obtained by the Wilcoxon signed-rank test. After 4 weeks of YH1 treatment, there were significant reductions in glycated albumin, 2hPG, total cholesterol, HDL-C, LDL-C, triglycerides, and triglycerides/HDL-C. There was a trend of decreased ALT level after YH1 treatment, but the difference was not significant. ALT, alanine aminotransferase; BMI, body mass index; Cr, creatinine; DBP, diastolic blood pressure; DM, diabetes mellitus; FPG, fasting plasma glucose; HDL-C, high-density lipoprotein cholesterol; HOMA-IR, homeostatic model assessment of insulin resistance; HOMA-β, homeostatic model assessment of β cell function; HR, heart rate; LDL-C, low-density lipoprotein cholesterol; 2hPG, 2-h postprandial glucose; SBP, systolic blood pressure.

## Data Availability

The data presented in this study are available on request from the corresponding author.

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
