# Peer review of "Gut Microbiota and Bile Acids Mediate the Clinical Benefits of YH1 in Male Patients with Type 2 Diabetes Mellitus: A Pilot Observational Study"

_pharmaceutics, 2022, doi:10.3390/pharmaceutics14091857_

Round 1
Reviewer 1 Report
1. Authors should give more results about side effects of YH1 supplementation.
2. Please explain about the dose of YH1 which choosen in the clinical study.
3. Please recheck and correct some of typing errors e.g. statistically significant "p" in line 283, 390 and Table1

Author Response
Dear Editors and the Reviewer #1,
Thank you for giving us the second opportunity to revise the manuscript. We appreciate your insightful feedback and constructive suggestions on our manuscript (ID: pharmaceutics-1894473). The reviewer’s comments are highlighted in bold, and our point-by-point responses to the reviewer's comments are provided as follows. Please note that the line numbers listed in this letter refer to our revised manuscript with tracked changes and not to the original manuscript. Thank you for your time and consideration.
To the Reviewer #1:
- Authors should give more results about side effects of YH1 supplementation.
=> The authors thank the reviewer for this comment. Our previous randomized, double-blind, placebo-controlled trial [1] documented detailed side effects of YH1, mainly gastrointestinal (GI) adverse events including diarrhea, constipation, nausea, bloating, and gastroesophageal reflux disease. With regard to the side effects of YH1 in this study, we have added lines 375-381 in the revised Results section and lines 487-488 in the revised Discussions section as follows.
From line 375, "Adverse GI event including mild nausea or bloating was found once in two patients, respectively, and the rest of 8 patients did not have any side effect during this trial. The severity of above GI events was below grade 2 as measured by the Common Terminology Criteria for Adverse Events (CTCAE, Version 4.0) grading system. YH1 was withdrawn temporarily for one day when these two participants reported discomfort, and the treatment was resumed after the symptoms resolved without other medical intervention. No patients experienced hypoglycemia during this study. "
From line 487, "Similarly, the incidence of adverse GI events, including nausea or bloating, in the YH1 group of previous trial and current study were 21.7% and 20%, respectively."
[1] Huang, Y.H.; Chen, S.T.; Liu, F.H.; Hsieh, S.H.; Lin, C.H.; Liou, M.J.; Wang, C.C.; Huang, C.H.; Liu, G.H.; Lin, J.R.; et al. The efficacy and safety of concentrated herbal extract granules, YH1, as an add-on medication in poorly controlled type 2 diabetes: A randomized, double-blind, placebo-controlled pilot trial. PLoS One 2019, 14, e0221199, doi:10.1371/journal.pone.0221199.
- Please explain about the dose of YH1 which chosen in the clinical study.
=> Our previous hospital-based retrospective cohort study [2] identified the effective hypoglycemic does of Rhizoma Coptidis extract granules for patients with type 2 diabetes. In addition, our previous randomized, double-blind, placebo-controlled pilot study [1] discovered that patients taking 6 g of YH1 (3 g of Rhizoma Coptidis and 3 g of Shen-Ling-Bai-Zhu-San) three times daily significantly improved hyperglycemia and hyperlipidemia in overweight/obese patients with poorly controlled type 2 diabetes who had taken three or more classes of oral hypoglycemic agents. To evaluate the possible mechanisms of YH1-mediated hypoglycemic and hypolipidemic effects, we chose the same dose of YH1 as in the previous clinical trial for this study.
[1] Huang, Y.H.; Chen, S.T.; Liu, F.H.; Hsieh, S.H.; Lin, C.H.; Liou, M.J.; Wang, C.C.; Huang, C.H.; Liu, G.H.; Lin, J.R.; et al. The efficacy and safety of concentrated herbal extract granules, YH1, as an add-on medication in poorly controlled type 2 diabetes: A randomized, double-blind, placebo-controlled pilot trial. PLoS One 2019, 14, e0221199, doi:10.1371/journal.pone.0221199.
[2] Huang, Y.H.; Liu, G.H.; Hsu, T.Y.; Yang, L.Y.; Lee, M.C.; Huang, C.T.; Wu, Y.H. Effective Dose of Rhizoma Coptidis Extract Granules for Type 2 Diabetes Treatment: A Hospital-Based Retrospective Cohort Study. Front. Pharmacol. 2020, 11, 597703, doi:10.3389/fphar.2020.597703.
- Please recheck and correct some of typing errors e.g. statistically significant "p" in line 283, 390 and Table1
=> We apologize for the confusion. Perhaps due to the system problems, many italicized characters in the original drafts have become regular characters. The American Psychological Association (APA) suggest "p value". The p is lowercase and italicized without hyphen between "p" and "value". Therefore, we have re-edited and revised all "p" value into "p" value.
We have responded to each question above and revised the manuscript accordingly. Please let us know if you have any additional questions and suggestions. We appreciate your time and effort. Thank you very much.
Sincerely yours,
Yueh-Hsiang Huang, MD PhD

Reviewer 2 Report
In this pilot study, Huang et al. provide a potential role of an herbal extract formula (YH1) composed of Rhizoma Coptidis and Shen-Ling-Bai-Zhu-San in 10 male patients with T2DM. Globally, this is an interesting and meritorious study demonstrating potential antidiabetic and metabolic effects of YH1 mediated by the gut microbiota, specially regarding fecal and plasma bile acids. The manuscript is properly written. However, i have some concerns that should be addressed before further processing of the manuscript.
Major:
Throughout the manuscript, the description of Rhizoma Coptidis is quite adequate, giving a central role of berberine as its main bioactive compound. The authors properly compared the benefits from using the entire herbal extract in comparison to the berberine alone, which makes sense regarding the available data. However, i do not understand at all the benefits and composition of Shen-Ling-Bai-Zhu-San. Appart from their effects on SCFAs producing bacteria, does it have any additional actions in the gut microbiota or bile acids? And is there any study comparing the action of YH1 versus Rhizoma Coptidis alone?
In the present study, a standarized breakfast was given to the participants in order to prevent possible bias. But how was this standarized breakfast? Was it with low content of fat? I feel that this standarized breakfast should be more detailed to deeper understand the design and effects of YH1
Growing evidence has revealed notable differences of BA and gut microbiota depending on male/female gender (see https://www.ncbi.nlm.nih.gov/pmc/articles/PMC5431816/). Despite i understand that the authors do not include women in this study due to methodological limitations and lack of scientific knowledge in this field, i feel that this fact should be reflected in the tittle of the present manuscript, in order to better represent the content of the present work ( Gut microbiota and bile acids mediate the clinical benefits of YH1 in male patients/men with type 2 diabetes mellitus: A pilot observational study). Besides, i would strongly recommend for future studies to include women population alone or with men to compare the results and add further knowledge in this relevant field.
Minor:
Please, review the use of italics when needed (i.e. in vivo and in vitro)
The references [14], [16], [17] should be written in the proper format
Author Response
Dear Editors and the Reviewer #2,
Thank you for giving us the second opportunity to revise the manuscript. We appreciate your insightful feedback and constructive suggestions on our manuscript (ID: pharmaceutics-1894473). The reviewer’s comments are highlighted in bold, and our point-by-point responses to the reviewer's comments are provided as follows. Please note that the line numbers listed in this letter refer to our revised manuscript with tracked changes and not to the original manuscript. Thank you for your time and consideration.
To the Reviewer #2:
- Throughout the manuscript, the description of Rhizoma Coptidisis quite adequate, giving a central role of berberine as its main bioactive compound. The authors properly compared the benefits from using the entire herbal extract in comparison to the berberine alone, which makes sense regarding the available data. However, I do not understand at all the benefits and composition of Shen-Ling-Bai-Zhu-San. Apart from their effects on SCFAs producing bacteria, does it have any additional actions in the gut microbiota or bile acids? And is there any study comparing the action of YH1 versus Rhizoma Coptidis alone?
=> We appreciate your feedback. YH1 contains Rhizoma Coptidis (RC) and Shen-Ling-Bai-Zhu-San (SLBZS), and the dose as well as composition including HPLC profiles of YH1 has been described in our previous clinical study [1] as we cited the reference in line 51 and line 121. In our previous study, we have discussed the advantages of SLBZS plus RC as YH1 formula, and the following sentences were mentioned. "Radix Ginseng, Rhizoma Dioscoreae, Rhizoma Atractylodis macrocephalae, and Poria are four major herbs of Shen-Ling-Bai-Zhu-San (SLBZS). These herbs exert antidiabetic effects in vitro and in vivo by enhancing insulin production/secretion, modulating antioxidant activities and inflammatory pathways, promoting the release of GLP-1, increasing glucose metabolism/uptake, or improving energy metabolism in skeletal muscle [2-5]. Therefore, Rhizoma Coptidis and SLBZS together should provide additive effects in treating poorly controlled type 2 diabetes." "SLBZS was reported to modulate the gut microbiota during the alleviation of antibiotic-associated diarrhea by enriching SCFA-producing bacteria[6]. Atractylenolide III, found in SLBZS, was noted to have a gastroprotective effect via inhibition of the matrix metalloproteinase (MMP)-2 and MMP-9 pathways [7]. Therefore, YH1 containing SLBZS could alleviate the negative GI effects of berberine from Rhizoma Coptidis." Currently, there was no clinical study mentioned about the effect of RC or SLBZS treatment alone on gut microbiota or bile acid metabolism for type 2 diabetes. Our previous hospital-based retrospective cohort study [8] identified the effective hypoglycemic does of RC extract granules for patients with type 2 diabetes, and RC plus SLBZS was the most prescribed herbal formula (77.0 % of all prescriptions) in clinical practice. Actually, Chinese medical doctors always prescribe formula rather than a single herb such as RC alone to treat diabetes based on the herb compatibility of traditional Chinese medicine. Therefore, we don’t have the opportunity to compare the action of YH1 versus Rhizoma Coptidis alone.
Reference:
- Huang, Y.H.; Chen, S.T.; Liu, F.H.; Hsieh, S.H.; Lin, C.H.; Liou, M.J.; Wang, C.C.; Huang, C.H.; Liu, G.H.; Lin, J.R.; et al. The efficacy and safety of concentrated herbal extract granules, YH1, as an add-on medication in poorly controlled type 2 diabetes: A randomized, double-blind, placebo-controlled pilot trial. PLoS One 2019, 14, e0221199, doi:10.1371/journal.pone.0221199.
- Shishtar, E.; Sievenpiper, J.L.; Djedovic, V.; Cozma, A.I.; Ha, V.; Jayalath, V.H.; Jenkins, D.J.A.; Meija, S.B.; de Souza, R.J.; Jovanovski, E.; et al. The Effect of Ginseng (The Genus Panax) on Glycemic Control: A Systematic Review and Meta-Analysis of Randomized Controlled Clinical Trials. PLoS One 2014, 9, e107391, doi:10.1371/journal.pone.0107391.
- Go HK, R.M., Kim GB, Na CS, Song CH, Kim JS, Kim SJ, Kang HS. Antidiabetic Effects of Yam (Dioscorea batatas) and Its Active Constituent, Allantoin, in a Rat Model of Streptozotocin-Induced Diabetes. Nutrients 2015, 7, 8532-8544, doi:10.3390/nu7105411.
- Song, M.Y.; Jung, H.W.; Kang, S.Y.; Park, Y.-K. Atractylenolide III Enhances Energy Metabolism by Increasing the SIRT-1 and PGC1α Expression with AMPK Phosphorylation in C2C12 Mouse Skeletal Muscle Cells. Biological and Pharmaceutical Bulletin 2017, 40, 339-344, doi:10.1248/bpb.b16-00853.
- Li TH, H.C., Chang CLT, Yang WC. Anti-Hyperglycemic Properties of Crude Extract and Triterpenes from Poria cocos. Evidence-based Complementary and Alternative Medicine : eCAM 2011, 2011, 128402, doi:10.1155/2011/128402.
- Lv, W.; Liu, C.; Ye, C.; Sun, J.; Tan, X.; Zhang, C.; Qu, Q.; Shi, D.; Guo, S. Structural modulation of gut microbiota during alleviation of antibiotic-associated diarrhea with herbal formula. International Journal of Biological Macromolecules 2017, 105, 1622-1629, doi:https://doi.org/10.1016/j.ijbiomac.2017.02.060.
- Wang, K.T.; Chen, L.G.; Wu, C.H.; Chang, C.C.; Wang, C.C. Gastroprotective activity of atractylenolide III from Atractylodes ovata on ethanol‐induced gastric ulcer in vitro and in vivo. Journal of Pharmacy and Pharmacology 2010, 62, 381-388, doi:10.1211/jpp.62.03.0014.
- Huang, Y.H.; Liu, G.H.; Hsu, T.Y.; Yang, L.Y.; Lee, M.C.; Huang, C.T.; Wu, Y.H. Effective Dose of Rhizoma Coptidis Extract Granules for Type 2 Diabetes Treatment: A Hospital-Based Retrospective Cohort Study. Front. Pharmacol. 2020, 11, 597703, doi:10.3389/fphar.2020.597703.
- In the present study, a standardized breakfast was given to the participants in order to prevent possible bias. But how was this standardized breakfast? Was it with low content of fat? I feel that this standardized breakfast should be more detailed to deeper understand the design and effects of YH1
=> The authors thank the reviewer for this comment. Indeed, diet is an important confounding factor that affects the 2-hour postprandial glucose and bile acid secretion of patients. Therefore, "a standardized breakfast was provided in this study to prevent different diet-related BA secretion" (lines 506-507). We have added lines 154-157 in the revised Method section to identify the details of the standardized breakfast as follows. From line 154, "A standardized breakfast containing a Chinese omelet with pork chop and a slice of pan-fried radish cake (total 440 calories, including 45 g carbohydrates, 20 g protein, and 20 g fat) was provided at 0 and 4 weeks to avoid the effect of different foods on BA profiles." This standardized breakfast was not a low-fat diet.
- Growing evidence has revealed notable differences of BA and gut microbiota depending on male/female gender (see https://www.ncbi.nlm.nih.gov/pmc/articles/PMC5431816/). Despite I understand that the authors do not include women in this study due to methodological limitations and lack of scientific knowledge in this field, I feel that this fact should be reflected in the tittle of the present manuscript, in order to better represent the content of the present work (Gut microbiota and bile acids mediate the clinical benefits of YH1 in male patients/menwith type 2 diabetes mellitus: A pilot observational study). Besides, I would strongly recommend for future studies to include women population alone or with men to compare the results and add further knowledge in this relevant field.
=> We understood the sex differences of BA and gut microbiota as mentioned in lines 504-506 "Females were excluded to avoid blood contamination of fecal samples during the menstrual period and to avoid sex differences in the microbiota and BA compositions." The authors thank the reviewer for this valuable comment, and we have made the suggested revision in lines 511-513 "Future research to enroll women together with men can determine if the influences of YH1 on gut microbiota and BA are universal or gender specific for patients with T2D and hyperlipidemia." In addition, the title of the manuscript was changed to "Gut microbiota and bile acids mediate the clinical benefits of YH1 in male patients with type 2 diabetes mellitus: A pilot observational study".
- Please, review the use of italics when needed (i.e. in vivoand in vitro)
=> Perhaps due to the system problems, many italicized characters in the original drafts have become regular characters. Therefore, we have re-edited and revised the manuscript, including italicizing "in vivo and in vitro" in line 62. Thank you for the friendly reminder.
- The references [14], [16], [17] should be written in the proper format
=> We appreciate this comment. The references [14], [16], [17] were revised in the proper format as follows.
- Panzitt, K.; Zollner, G.; Marschall, H.U.; Wagner, M. Recent advances on FXR-targeting therapeutics. Mol. Cell. Endocrinol. 2022, 552, 111678, doi:10.1016/j.mce.2022.111678.
- Duboc, H.; Taché, Y.; Hofmann, A.F. The bile acid TGR5 membrane receptor: from basic research to clinical application. Dig. Liver Dis. 2014, 46, 302-312, doi:10.1016/j.dld.2013.10.021.
- Sepe, V.; Festa, C.; Renga, B.; Carino, A.; Cipriani, S.; Finamore, C.; Masullo, D.; Del Gaudio, F.; Monti, M.C.; Fiorucci, S.; et al. Insights on FXR selective modulation. Speculation on bile acid chemical space in the discovery of potent and selective agonists. Sci. Rep. 2016, 6, 19008, doi:10.1038/srep1900
We have responded to each question above and revised the manuscript accordingly. Please let us know if you have any additional questions and suggestions. We thank you for the valuable comments and appreciate your time and effort.
Sincerely yours,
Yueh-Hsiang Huang, MD PhD
